# Numerical Analysis of the Heterogeneity Effect on Electroosmotic Micromixers Based on the Standard Deviation of Concentration and Mixing Entropy Index

**DOI:** 10.3390/mi12091055

**Published:** 2021-08-31

**Authors:** Alireza Farahinia, Jafar Jamaati, Hamid Niazmand, Wenjun Zhang

**Affiliations:** 1Department of Mechanical Engineering, University of Saskatchewan, Saskatoon, SK S7N 5A9, Canada; chris.zhang@usask.ca; 2Department of Mechanical Engineering, Razi University, Kermanshah 6714414971, Iran; j.jamaati@razi.ac.ir; 3Department of Mechanical Engineering, Ferdowsi University of Mashhad, Mashhad 9177948974, Iran; niazmand@um.ac.ir

**Keywords:** mixing, electroosmotic micromixer, Helmholtz–Smoluchowski model, heterogeneous zeta-potential

## Abstract

One approach to achieve a homogeneous mixture in microfluidic systems in the quickest time and shortest possible length is to employ electroosmotic flow characteristics with heterogeneous surface properties. Mixing using electroosmotic flow inside microchannels with homogeneous walls is done primarily under the influence of molecular diffusion, which is not strong enough to mix the fluids thoroughly. However, surface chemistry technology can help create desired patterns on microchannel walls to generate significant rotational currents and improve mixing efficiency remarkably. This study analyzes the function of a heterogeneous zeta-potential patch located on a microchannel wall in creating mixing inside a microchannel affected by electroosmotic flow and determines the optimal length to achieve the desired mixing rate. The approximate Helmholtz–Smoluchowski model is suggested to reduce computational costs and simplify the solving process. The results show that the heterogeneity length and location of the zeta-potential patch affect the final mixing proficiency. It was also observed that the slip coefficient on the wall has a more significant effect than the Reynolds number change on improving the mixing efficiency of electroosmotic micromixers, benefiting the heterogeneous distribution of zeta-potential. In addition, using a channel with a heterogeneous zeta-potential patch covered by a slip surface did not lead to an adequate mixing in low Reynolds numbers. Therefore, a homogeneous channel without any heterogeneity would be a priority in such a range of Reynolds numbers. However, increasing the Reynolds number and the presence of a slip coefficient on the heterogeneous channel wall enhances the mixing efficiency relative to the homogeneous one. It should be noted, though, that increasing the slip coefficient will make the mixing efficiency decrease sharply in any situation, especially in high Reynolds numbers.

## 1. Introduction

Microtools and microfluidics have been used in many engineering fields in recent years [1,2,3,4], and the fluid flow pattern and how it is mixed has received much attention [5,6,7]. The most critical point about mixing on a micron-scale is that the cause of this phenomenon is molecular diffusion, which is an inherently slow process [8,9]. Turbulence and creating disturbances can dramatically increase the mixing rate in macro-scale flows; however, creating turbulent flows in microchannels requires a high-pressure drop within the microchannel [10,11,12]. Hence, the flow inside the straight or even curved microchannels is in the laminar flow range in most practical applications. For this reason, mixing enhancement inside the microchannel through the generation of turbulence does not seem reasonable.

Mixing inside a microchannel can be done in two ways: passive or active, each of which has proposed different techniques [13,14,15]. Passive micromixers provide favorable mixing only through their inherent structure and thus, have many advantages, such as low cost, ease of construction, and no need for moving parts or additional power [16,17]. The fluid flow is forced to change direction, split, combine, or develop the contact surface in such passive structures [18,19,20,21]. An external energy source accompanies the required energy for driving fluid in active mixing techniques, which dramatically improves the mixing efficiency [22,23].

One of the most effective methods for mixing in a microchannel is electroosmotic flow, in which the ion-containing fluid has to move near the surface under an applied electric field [24,25]. The electroosmotic flow field is highly dependent on the electrochemical properties of the microchannel wall surface and the fluid. The electroosmotic flows are complicated and have rotational regions in microchannels whose wall properties, especially the wall zeta-potential amount, are heterogeneous [26]. The benefits of active and passive methods can thus be enjoyed simultaneously in these flows. The simplest case for a heterogeneous microchannel is that a segment of its wall’s properties is locally different from other points, while the other sections have similar characteristics along the homogeneous wall. This non-homogeneous section is called a patch. Figure 1 qualitatively illustrates the electroosmotic flow inside such a microchannel. The wall zeta-potential is considered negative for the heterogeneity segment, so there is positive zeta-potential everywhere except for the heterogeneity patch.

Microchannels can be designed through proper heterogeneity arrangement on channel surfaces for mixing, pumping, or both [27,28]. Under such a circumstance, the numerical solution of electroosmotic flow in heterogeneous channels without using the Helmholtz–Smoluchowski (H–S) approximate model is time-consuming [29,30], and few studies have been done in this field. Generally, simulating the velocity field in complex geometries inside a channel with heterogeneous walls requires the simultaneous solving of the electric field, the charge distribution, and the Navier–Stokes equation [31,32].

On the other hand, the H–S approximate model simplifies the modeling of electroosmotic flows because it solves the velocity field and electric field independently [29,30]. The H–S condition is applied to the boundaries with this simplification, and the velocity field within the flow domain is then solved using this boundary condition. The H–S model’s validity has been well studied for investigating the fluid flow field [33], and determining concentration field and mixing efficiency has shown that this model can have high accuracy for predicting mixing under suitable conditions and that the maximum error is negligible [34].

The surface-to-volume ratio is high in microchannels due to the small length scale, so surface phenomena, useless in large-scale flows, are crucial in such microfluidic devices. One of these surface phenomena playing an essential role in microfluidics is the slip condition of fluid on the surface. Experimental studies have shown that, in liquid flows, even at low Reynolds numbers (Re < 10), significant slip occurs on walls having low energy levels (hydrophobic) [35,36]. The slip condition on rough and conventional walls is attributed to the presence of nanobubbles trapped in surface cavities [37]. It has been confirmed that bubbles are well-formed on the hydrophobic surfaces as well [38]. Therefore, nanobubbles are one reason for the slip both on perfectly polished surfaces and on unsmooth surfaces. Molecular dynamics simulations of the water–solid interface have shown that, in the presence of hydrodynamic slip, the distribution of electric charges is correctly estimated by the Poisson–Boltzmann equation [39]. It seems that the presence of surface charges causes more wettability and reduces the slip of fluid on the surface; however, experimental observations have shown that significant slip occurs on highly charged surfaces [40]. Besides, the experimental evidence shows that the presence of slip significantly enhances the amount of wall zeta-potential [41]. It has been claimed that the presence of slip amplifies the zeta-potential with a coefficient of (1 + Kβ). Considering the typical slip coefficient (β) values, zeta-potential has a significant increase for large values of the dimensionless Debye–Huckel parameter (K). A presented theoretical model based on the free energy of binary mixtures confirms it [42].

The effects of slip on the fluid flow inside the microchannel have also been investigated and found that the presence of slip in electroosmotic flows increases the fluid’s mass flow rate and significantly decreases the required applied voltage [40]. The electroosmotic flow inside the hydrophobic microchannel has been studied using experimental relationships to determine the slip velocity on the walls [43]. Finally, simultaneously evaluating the zeta-potential and slip coefficient has been proposed [44].

The energy conversion efficiency of electrokinetic systems in nanochannels has been numerically studied by considering the slip effect and using elliptic functions for the potential field [45]. The slip presence led to a significant enhancement of energy conversion efficiency in a similar study [46]. Other studies in this field include the investigation of surface slip phenomenon in the electroosmotic flow of non-Newtonian fluids in porous media [47], the surface slip examination in microfluidic equipment [48], the analytical and numerical study of slip flow of Newtonian fluids inside flat microchannels [34,49], and analytical study of electroosmotic slip flow of viscoelastic fluids with pressure gradient in flat [50] and annular microchannels [51].

As can be seen from the above papers, the slip coefficient’s effect on mixing inside the microchannels has been less noticed. Therefore, in addition to a comprehensive and straightforward mixing analysis that can be generalized to other micromixers, this study assessed the use of an electroosmotic micromixer with a heterogeneous patch on the wall and investigated the role of this parameter in improving or reducing mixing efficiency.

The most common mixing criterion is the concentration-based index applied in both homogeneous and heterogeneous zeta-potential channels and provides acceptable results [18,52]. Another criterion used to study the mixing of microchannels is the mixing entropy criterion [34,53,54]. Unlike the previously introduced criterion, this criterion is mainly employed in channels without heterogeneity on the wall. This index can be used along with the concentration criterion so that the results, especially in three-dimensional analysis, can be well examined. The mixing rate evaluation is based on the weighted average of the species concentration’s standard deviation by applying the velocity weighting function and mixing entropy with the mass flow rate weighting function inside the mentioned microchannel with non-uniform zeta-potential on the wall through solving Nernst–Planck equations. Applying the slip coefficient has noticeable results that will be mentioned in the following sections.

Despite studies and investigations in the electroosmotic flow field and the examination of the intensity and size of heterogeneity, there is no discussion about the effect of the heterogeneous patch’s size and position to achieve a specific mixing efficiency along the optimal length of the microchannel. Therefore, this issue has been investigated using the approximate H–S model in the current paper to address this gap.

This study proposed a microfluidic method to produce micro-scaled turbulence by inserting zeta-potential heterogeneity on the microchannel walls and the underlying fluid mechanics. It should be noted that the mechanism of placing the heterogeneity patch on the microchannel wall to form vortices is out of the scope of this paper. However, this paper’s discussion includes the heterogeneity patch role in the mixing process, recognized in the electroosmotic micromixer. Related techniques such as numerical methods to model the vortex generation process and flow pattern observation are also presented.

This paper is structured into five main sections (besides this introduction section). The following section discusses the fluid mechanics governing the electroosmotic flow and the approximate model for simulating such a flow inside a microchannel. The Section 3 describes the numerical approach to solving the coupled equations and the approximate model. The Section 4, divided into three subsections, discusses the effect of heterogeneity location and its size on mixing efficiency. This section also briefly discusses slip and its impact on both homogenous and non-homogeneous microchannels compared to the Reynolds number variations. Conclusions, along with a discussion of the numerical model’s accuracy and the proficiency of mixing criteria, will be presented in the Section 5.

## 2. Governing Equations

Dielectric surfaces store an electric charge when exposed to an electrolyte solution. The electric charge of the surface affects its adjacent ions in the solution, attracts unlike-charged ions, and repels the ions with the like charge. With the charged layer on the wall, this layer is called the electric double layer (EDL) [55]. Moreover, according to the static electricity theory, electric charges create an electric field inside the electric double layer. The amount of electric potential on the surface is also called the zeta-potential.

The following hypotheses have been considered in deriving the equations governing the electroosmotic flow:The solution is a Newtonian incompressible fluid.No chemical reaction occurs between the species or between the species and the microchannel wall.The electrolyte solution is symmetric, meaning that the positive and negative valence number (z) of ions is equal.Gravitational and buoyancy effects are negligible.The steady-state of the equations are extracted.

The governing equations introduced below include the Poisson equation for distributing the electric field inside the electric double layer, the Nernst–Planck equation for the concentration distributing of positive and negative ions, the modified Navier–Stokes equation for the velocity field, and the concentration equation.

### 2.1. Poisson–Boltzmann Equation

Based on the electrostatic theory, the relationship between the distribution of electric potential and that of electric charges is determined by the Poisson–Boltzmann equation; the dimensionless form of which is shown in Equation (1) [56]:(1)$$\nabla^{2}\left(\psi +\Phi \right)=\frac{K^{2}}{2}\left(n^{+}-n^{-}\right)$$
where ψ is the distribution of electric potential due to electric double layer; Φ is the distribution of external electrical potential because of electrodes; $n^{+}$ shows a numerical concentration of positive ions; $n^{-}$ illustrates a numerical concentration of negative ions, and K=κH is the dimensionless form of the Debye–Huckel parameter. 

$\kappa^{-1}={\left({\varepsilon K}_{b}T/2n_{0}z^{2}e^{2}\right)}^{1/2}$ determines the characteristic thickness of the electric double layer and z, e, $n_{0}$, ε, $K_{b}$, and T are the symmetric electrolyte valence number, the base charge of the electron, the numerical concentration of the ion mass in the uniform solution, the dielectric constant of the solution, the Boltzmann constant, and the absolute temperature of the electrolyte, respectively. 

### 2.2. Nernst–Planck Equations

The calculation of ionic concentrations is obtained by solving the ion (a charged chemical species) movement equations known as the Nernst–Planck equations. The dimensionless form of these equations is shown in Equations (2) and (3) [56]:(2)∇→·(V→n+)={∇2n++∇→·[n+(∇→ψ+A∇→Φ)]}Re Sc+
(3)∇→·(V→n−)={∇2n−+∇→·[n−(∇→ψ+A∇→Φ)]}Re Sc−

$\vec{V}$, $Sc^{\overset{+}{-}}=\mu /\rho D^{\overset{+}{-}}\;$, and Re are velocity vector, Schmidt number, and Reynolds number based on the reference velocity in the electroosmotic flow $U_{\mathrm{ref}}={\varepsilon E}_{\mathrm{ref}}\zeta /\mu$, respectively. $D^{+}$, and $D^{+}$ are molecular diffusion coefficients for positive and negative ions, respectively. 

$A=E_{\mathrm{ref}}H/(K_{b}T/\mathrm{ze})$ is a dimensionless parameter representing the ratio of the externally applied voltage to the basis voltage. In Equations (2) and (3), the parameter on the left of the equations is related to the displacement of ions; the first parameter on the equation’s right side is related to the molecular diffusion of ions, and the next is a dispersion due to ion movement, originated from electrical potential.

### 2.3. Navier–Stokes Equations

The velocity equations in a system under electrokinetic effects for fluid flow field under steady conditions with constant physical properties are written in the dimensionless Equation (4) [56]:(4)∇→(V→·V→)=−∇→P+1Re∇2V→−Bρe(∇→ψ+A∇→Φ)

$B=n_{0}K_{b}T/{\rho U}_{\mathrm{ref}}^{2}$ is a dimensionless parameter representing the ratio of ionic pressure to dynamic pressure; the dimensionless pressure is defined as $P=P^{*}/{\rho U}_{\mathrm{ref}}^{2}$. The last term in Equation (4) is the volumetric force due to the electric field’s effects and the fluid’s charged ions. This electric force acts like a gravitational force as a volumetric force and causes the fluid to move in electroosmotic flows.

### 2.4. Species Concentration Equation

The scalar field for the concentration of a species must be solved to investigate the mixing phenomenon. The equation governing the concentration field is in the form of Equation (5) [57]:(5)V→·∇→C=1Re Sc∇2C
where Sc=μ/ρD is the Schmidt number for the species, and Re is the Reynolds number. The concentration distribution at the microchannel inlet is as follows (Equation (6)):(6)C(x=0, y)={0             0<y<H2 1           H2<y<H

### 2.5. Helmholtz–Smoluchowski Modeling

A suitable boundary condition can model all effects related to electrical charges and the electric field for the momentum equation under particular circumstances, such as the uniformity of zeta-potential along the wall, the thinness of the electric double layer compared to the channel width, non-conductivity of the channel walls, low Reynolds number, and the uniformity of properties and electrolyte temperature. This slip boundary condition is determined based on the applied electric field and the size of charge on the wall in Equation (7) [57,58]:(7)$$u_{s}=-\frac{{\varepsilon E}_{\mathrm{ext}}}{\mu }\zeta$$
where $u_{s}$ is the slip velocity amount of the fluid adjacent to the wall (or to better say, at the electric double layer edge); ε is the solution dielectric constant; $E_{\mathrm{ext}}$ is the external field strength; ζ is the wall zeta-potential value, and μ represents the fluid viscosity.

Applying the above condition leads to converting the electric double layer’s effect and the presence of ions on the fluid flow as a slip boundary condition on the solid wall. In this way, the effect of electric force in the Navier–Stokes equation is applied through the slip condition. The electroosmotic velocity field is solved only by solving the Navier–Stokes equations with the mentioned slip condition, without considering the volumetric force and without solving the electric charge and potential fields. The Equation (8) shows the final equation after applying these assumptions [57,58]:(8)∇→(V→·V→)=−∇→P+1Re∇2V→

Therefore, the mixing simulation under electroosmotic flow is obtained through H–S modeling and solving Equations (7) and (8). In contrast, full modeling of such a flow is too complicated without this simplification because it involves solving several coupled Equations (1)–(5).

### 2.6. Concentration Evaluation Criteria

A suitable parameter is required to determine the mixing rate to evaluate and compare different performances of micromixers. The mixing efficiency for a micromixer is introduced as Equation (9) [33]:(9)$$\varepsilon_{m}=\frac{1-\sigma_{w}}{1-\sigma_{w,\;\min}}$$

$\sigma_{w}$ is the standard deviation of the concentrations in each cross-section, representing the non-uniformity of the concentration in that section, which is defined as Equation (10):(10)$$\sigma_{w}^{2}=\frac{\int_{0}^{H}u\left(y\right){\left(C\left(y\right)-C_{m}\right)}^{2}\mathrm{dy}}{\int_{0}^{H}u\left(y\right)\mathrm{dy}}$$

In Equation (10), calculating the mixing efficiency, the effects of fluid rotation on the mixing are performed by applying the velocity weight function, while analyzing the mixing rate in complex flows. H is the channel height, and $C_{m}$ is the mean concentration at each cross-section calculated by Equation (11):(11)$$C_{m}=\frac{\int_{0}^{H}u\left(y\right)C\left(y\right)\mathrm{dy}}{\int_{0}^{H}u\left(y\right)\mathrm{dy}}$$

### 2.7. Mixing Entropy Index

Entropy can also be used as a measure to evaluate mixing. Shannon’s entropy criterion has been proposed based on concentration values in discrete points. Unlike the concentration deviation criterion in which the weight function was applied; this time, the mass flow weighting function is applied to prevent non-physical fluctuations in the mixing efficiency amount and determine the increase or decrease in mixing. The modified criterion is defined as a Equation (12) [54]:(12)$$S_{\mathrm{mix}}=\frac{-\int_{A}^{\;}C_{\;}\mathrm{Ln}\left(C\right)\;\rho u\;\mathrm{dy}}{\int_{A}^{}\rho u\;\mathrm{dy}}$$

The mixing efficiency is defined as Equation (13) based on using the weighted entropy criterion at each microchannel section.
(13)$$\varepsilon_{s}=\frac{S_{\mathrm{mix}}-S_{\mathrm{inlet}}}{S_{\infty }-S_{\mathrm{inlet}}}$$

For the introduced channel, according to the microchannel inlet’s concentration distribution, if the channel length is long enough, complete mixing will occur so that the final concentration value will be 0.5. For this case study, the entropy index value is equal to its maximum amount, $S_{\infty }=-0.5\;\mathrm{Ln}\left(0.5\right)$. In addition, the entropy index value is equal to its minimum (zero) at the inlet. Therefore, the mixing efficiency is calculated according to Equation (13) as $\varepsilon_{s}=S_{\mathrm{mix}}/0.347$.

## 3. Numerical Method and Validation

The external electric field, Φ, is first obtained to solve the equations numerically. Then, the potential value originating from the electric double layer (i.e., ψ) inside the region is equal to zero as a first guess and is equal to the zeta-potential amount on the walls. Next, the equations associated with the internal and external electric fields are solved with the initial values $n^{+}=n^{-}=0$, and the internal potential field, ψ, is developed. Nernst–Planck equations are then solved to obtain the concentration distribution for the positive and negative ions, $n^{+}$ and $n^{-}$. Then, the density of electric charges $\rho_{e}$ is obtained from the relation $\rho_{e}=n^{+}-n^{-}$. The first estimate for volumetric electric force can be calculated at this stage, which can be used to obtain the velocity field.

The pressure field is first guessed to calculate the flow field, and the momentum equations are then solved to achieve the velocity field by employing the finite volume method with homotopic variables in a non-uniform network. The SIMPLE method determines the relationship between velocity and pressure field. The Ray-Chou intermediate scheme is also used to calculate the convection mass flow rate to avoid the non-physical fluctuations in the flow field [59]. The continuity equation is then solved using the obtained velocity field to modify the pressure and velocity fields. Since the flow field affects the electric charge density, the Nernst–Planck equations are solved again to obtain the distribution of ionic concentrations and the net density of the electric charges. Subsequently, the distribution of external and internal potential is calculated. These steps are repeated until appropriate convergence is achieved. The concentration equation is solved after the velocity equation converges.

The electroosmotic flow field in a heterogeneous microchannel with the length of L is solved to validate the H–S approximate model. The zeta-potential value on the upper wall is equal to $\zeta^{t}\left(x\right)/\zeta_{0}=1+4\sin\left(2\pi x/L\right)$ and on the lower wall is equal to $\zeta^{b}\left(x\right)/\zeta_{0}=1+4\sin\left(4\pi x/L\right)$. According to Equation (7), slip velocities are generated heterogeneously due to the heterogeneous distribution of surface charges on the wall. Under such conditions and based on the H–S model, the slip boundary condition is $u^{t}\left(x\right)/u_{0}=1+4\sin\left(2\pi x/L\right)$ on the upper wall and $u^{b}\left(x\right)/u_{0}=1+4\sin\left(4\pi x/L\right)$ on the lower wall. The numerical results of the flow field using a uniform network with 80 × 330 computational nodes are compared with the flow field obtained from the analytical solution [60] in Figure 2.

For example, the position of the vortex center in Figure 2, the point N, is calculated by the analytical solution N (x, y) = (6.5, 0.75), and the position of the single point, M, is equal to M(x,y) = (5.5, 1.22). The positions of these points are obtained by numerical solution of the present paper equal to N(x,y)= (6.2,0.75) and M(x,y) = (4.0,1.20). The maximum relative error in predicting the location of single points for this complex flow field is equal to ∆y/H=0.01 in the transverse direction (y-direction) and ∆x/L=0.025 in the longitudinal direction.

The velocity equation for a steady, fully developed flow inside a flat microchannel will be as a Equation (14):(14)$$\mu \frac{\partial^{2}u^{\;}}{\partial y^{2}}+\rho_{e}^{\;}E_{x}^{\;}=0$$

The momentum equation after substituting the electric charge density based on the electric field is shown in Equation (15):(15)$$\frac{\partial^{2}u}{\partial y^{2}}=\frac{2E_{x}}{K^{2}}\mathrm{BRe}\;\nabla^{2}\Psi$$
where $B=n_{0}K_{b}T/{\rho U}_{\mathrm{ref}}^{2}$ is a dimensionless parameter representing the ratio of ionic pressure to dynamic pressure, and Re=ρUrefH/μ is the Reynolds number. Equation (15) has become dimensionless based on the reference velocity in the electroosmotic flow (i.e., $U_{\mathrm{ref}}={\varepsilon E}_{\mathrm{ref}}\zeta /\mu$).

The electric potential distribution, Ψ, is obtained by employing the Debye–Huckel approximation and solving the Poisson–Boltzmann equation. After substituting it in Equation (15), considering the slip boundary condition on the walls (u=βdu/dy) and applying the symmetry condition in the microchannel center (du/dy=0), the Equation (16) is achieved for the velocity profile: (16)$$\frac{u\left(y\right)}{U_{\mathrm{ref}}}=\left[1+K\beta \tanh\left(K/2\right)-\frac{\cosh\left(\mathrm{Ky}-K/2\right)}{\cosh\left(K/2\right)}\right]$$

Equations (17) and (18) show the analytical solution for the electric potential and velocity distribution (without considering the slip) [16,61]:(17)$$\frac{u\left(y\right)}{U_{\mathrm{ref}}}=1-\frac{\psi \left(y\right)}{\zeta }$$
(18)$$\frac{\psi \left(y\right)}{\zeta }=\frac{\cosh\left(\mathrm{Ky}-K/2\right)}{\cosh\left(K/2\right)}$$

After substituting the above relations in Equation (16), the velocity distribution in the presence of slip is acquired as Equation (19):(19)$$\frac{u\left(y\right)}{U_{\mathrm{ref}}}=\left[1+K\beta \tanh\left(K/2\right)-\frac{\psi \left(y\right)}{\zeta }\right]$$

The numerical results are compared with the analytical results of solving an ideal electroosmotic flow between two flat homogeneous channels without the presence of slip conditions on the walls to validate the numerical procedure. On the other hand, Figure 3 is drawn to validate and verify the numerical solution in the slip condition presence and to compare it with the analytical solution of Equation (15) [34,62]. Results are reported for two different slip coefficient values. As can be inferred, the acceptable consistency of the two solutions with or without the presence of slip condition on the walls indicates the high accuracy of the numerical solution in analyzing such surfaces.

## 4. Results

In this section, the results are divided into two parts. First, the size and the location of the heterogeneity patch on the wall are discussed using the introduced concentration criterion and the H–S approximate solution. In the second step, the results of applying the slip coefficient on the wall through the entropy mixing index employing the full numerical solution model will be examined.

### 4.1. The Effect of Heterogeneity Patch 

A hypothetical flow in the microchannel was studied, and the mixing rate of the species was calculated using the H–S approximate model and the appropriate slip boundary conditions. The slip boundary conditions on the upper and lower walls of the microchannel are considered Equations (20) and (21):(20)$$u_{b}\left(x\right)=\{\begin{array}{ll} 1, & 0<x\leq 2\\ 1+4\sin\left(\phi +\frac{6\pi x}{L}\right)\;\;, & 2<x\leq 4\\ 1, & 4<x\leq 6 \end{array}$$
(21)$$u_{t}\left(x\right)=u_{b}\left(x\right)$$

The concentration of studied species at the microchannel inlet is assumed to be zero in the lower half and equal to one in the upper half. The flow and concentration fields for Sc=50 and different angle values, ϕ, are shown in Figure 4. As can be seen qualitatively, the heterogeneous region (i.e., 2<x≤4) has significantly improved the mixing rate.

Figure 5 illustrates the calculated mixing efficiency for the velocity and concentration fields under Equations (20) and (21), for Sc=50, and different values of the angle ϕ.

For the examined diagrams in Figure 5, the mixing efficiency before the heterogeneous section (i.e., 2<x≤4) is almost the same for all microchannels. The significant difference occurs after the heterogeneity patch. This difference remains until the microchannel outlet; therefore, it can be inferred that the heterogeneous region has remarkably enhanced the mixing rate.

The mixing efficiency at the end of each microchannel describes that the performance of the microchannel in mixing can be used as a criterion for comparing two different micromixers’ proficiency. For example, according to Figure 5, the final mixing amount for a homogeneous microchannel reaches 91.9%, equal to 96% for the case (ϕ=π). 

At first glance, it may seem that this rate of improvement is not impressive. However, more attention reveals that the microchannel length should be significantly increased for a slight increase in mixing efficiency, especially at the high mixing efficiency values. For example, it can be seen from Figure 5 that a homogeneous microchannel at length equal to 6H brings the mixing efficiency to 91.9%, while the optimal microchannel (ϕ=π) at a shorter length, L≈4H, achieves such an efficiency, which is equivalent to a 30% reduction in length.

The mixing efficiency of the perfectly homogeneous microchannel can be a suitable reference for analyzing other microchannels. Figure 6 displays the final efficiency of different microchannels and their relative efficiency with respect to the homogeneous microchannel. According to Figure 6, it can be seen that, for the studied microchannels whose flow and concentration fields have been expressed in Equations (20) and (21), changing the heterogeneity pattern (through changing the phase of the sinusoidal function of the applied velocity to the wall) can improve the microchannel mixing by up to 4%.

Therefore, an important conclusion is that not only the position and size of the heterogeneity patch play a crucial role in optimizing the mixing but also that the zeta-potential distribution in the heterogeneity piece is essential. It should be noted again that this seemingly small improvement in microchannel mixing efficiency can lead to a significant reduction in its length and should therefore not be considered trivial.

Figure 7 shows the relationship between the microchannel mixing efficiency and its length more clearly. This figure investigates how much length of the microchannel is required to achieve a specific mixing efficiency. As a numerical example, it requires a length equal to 5.3 H to achieve a mixing efficiency of 90% in a homogeneous microchannel. In comparison, this mixing efficiency is achieved in length equal to 3.5 H of the optimal microchannel. This means that the length of the microchannel, in this case, can be reduced by 34%. In Figure 7, the length required for mixing consists of two sections: one homogeneous and the other heterogeneous.

Figure 8 examines the length required after the heterogeneity patch for a better conclusion. In other words, the performance of the heterogeneity patch itself is more meticulously investigated in this figure. This figure shows how much extra length ($L_{a}$) will be required to obtain the desired mixing efficiency after flow passing across the heterogeneous piece. Negative numbers are observed for the required length of the microchannel in some cases, indicating that the desired efficiency has been generated within the heterogeneous region, and there is no need for additional length after the heterogeneous piece.

#### 4.1.1. The Effect of Heterogeneity Position

The position effect of the heterogeneous patch on the mixing rate is shown in Figure 9. The calculated values for the mixing amount along the microchannel indicate that for a heterogeneous piece of a certain length, as the heterogeneity patch is closer to the microchannel inlet, the microchannel performs excellently to achieve better mixing. Therefore, inserting a heterogeneous patch at the beginning of the microchannel will significantly reduce the required microchannel length, especially in cases where the purpose of microchannel design is to achieve a specific mixing efficiency. However, placing a heterogeneous piece at the beginning of the microchannel causes the produced vortexes to affect the inlet flow so that backflows are generated at that area. Numerical modeling for such flow states requires more attention and consideration; in addition, real physical conditions must be noted so that modeling is not merely theoretical.

#### 4.1.2. The Effect of Heterogeneity Size 

It is observed that the mixing efficiency, calculated in each cross-section, always increases with the fluid movement along the microchannel so that it peaks at the end of the channel (x = L). The mixing efficiency at the microchannel outlet can be used as a suitable parameter in evaluating the micromixer. The relative mixing efficiency can also determine how much the designed patterns on the wall have improved the final mixing.

Figure 10 shows the mixing improvement by changing the size of the heterogeneity patch inside a microchannel having a heterogeneity piece in the lower wall. The value of the microchannel efficiency, $\epsilon_{L}$, and its relative efficiency are shown in this figure.

The relative efficiency is defined as the ratio of the mixing efficiency at the microchannel outlet (L) to the mixing efficiency at the uncharged (or even entirely homogeneous charged) microchannel output under the same conditions (i.e., similar pressure difference or potential difference).

It can be seen that increasing the length of the heterogeneity patch leads to improving the mixing efficiency by up to about 4%. According to the diagram gradient, the mixing growth rate is higher when the heterogeneity patches are smaller. A heterogeneous piece with a length $L_{\mathrm{patch}}=H$ (H is the microchannel height) on the channel wall increases the mixing rate by 2.5% compared to a completely homogeneous microchannel. This efficiency improvement for a heterogeneity piece with a length $L_{\mathrm{patch}}=2H$ on the same wall is equal to 4%. Therefore, it is appropriate to work with several smaller heterogeneity patches instead of inserting a large piece. The common point in all studied cases is that the presence of a heterogeneous patch on the wall and the generated vortexes results in improving the mixing rate in any location with any size.

### 4.2. The Effect of Slip on a Heterogeneity Patch

As the boundary condition u=βdu/dy, the fluid slip condition is applied to the walls to study the effect of surface hydrophobicity on mixing. Figure 11 depicts the mixing efficiency changes for different slip coefficients inside a channel with a heterogeneous patch based on the mixing entropy index. It is observed that mixing efficiency decreases when the slip coefficient increases. This is true because increasing β means facilitating the fluid flow moving through the microchannel; in other words, the pumping effect dominates the mixing effect.

Examining the previously mentioned heterogeneous microchannel flow field for the two different slip coefficients shown in Figure 12 confirms this finding. As the strength of the vortices declines at high slip coefficients, the mixing efficiency also decreases.

Figure 13 illustrates the mixing efficiency changes for the slip coefficient of 0.025 and different Reynolds numbers based on the mixing entropy index. It can be seen from this figure that the presence of slip in each Reynolds number reduces the mixing efficiency compared to when there is no slip. In this case, the general trend for mixing efficiency changes along the microchannel is like the non-slip mode, and the only difference is its size.

Examining efficiency at other slip values shows that the mixing efficiency experiences a greater decrease at higher slip coefficients. This issue is shown in Figure 14 in which the slip coefficient is set to 0.1 for the surface. Comparing Figure 13 and Figure 14 displays that increasing slip coefficient from 0.025 to 0.1 (i.e., by quadrupling) for Reynolds 0.001 causes the mixing efficiency to decrease from 79.3% to 61.1% (i.e., a reduction of about 18.2%), which is quite significant for the mixing efficiency. Similar results have been achieved through the standard deviation of concentration.

The mixing efficiency changes of heterogeneous microchannel for different slip coefficients for a specific Reynolds number based on the mixing entropy criterion are shown in Figure 15 and compared with the homogeneous channel. It can be seen in Figure 15 that the mixing efficiency continuously decreases with increasing slip coefficient for the shown arrangement and Re = 0.001. Under such circumstances, the mixing efficiency values at the microchannel output range from 61.1% to 77%. The efficiency amount of homogeneous microchannel at its outlet will be in the same range and approximately 83.1%.

It will be seen that such a situation will not continue as the Reynolds number increases. The diagram in Figure 16 is drawn with the same conditions as Figure 15, except that Re = 0.009 is assumed. The mixing efficiency values for heterogeneous microchannels at their outlets range from 52.6% to 59%, while the homogenous microchannel efficiency is about 31.6% in Figure 16. It means that the mixing efficiency changes a maximum of 18% with a 9-time enhancement of Reynolds number for the slip coefficient of 0.025. In contrast, it can be seen from Figure 13 and Figure 14 that a similar change for mixing efficiency has been obtained only by quadrupling the slip coefficient. Therefore, we conclude that the mixing of heterogeneous microchannels is affected more by the change in the slip coefficient amount than the Reynolds number change.

However, the mixing of homogeneous channels is strongly influenced by the Reynolds number so that their mixing efficiency will be drastically reduced with increasing Reynolds. Their amount will be even less than the efficiency value of heterogeneous channels with hydrophobic surfaces. This is quite different from the previous case (Figure 15), where it had an even better mixing efficiency than the channel having a heterogeneity patch on its wall with a slip coefficient.

Therefore, the wall’s hydrophobicity will not favorably affect the mixing efficiency in the range of low Reynolds numbers because the presence of slip even worsens its performance with respect to a homogeneous channel without heterogeneity patches (Figure 15). Conversely, suppose this type of channel will be used at high Reynolds numbers (Figure 16); in that case, it is better to apply the slip coefficient to the wall to achieve higher mixing efficiency than the homogeneous channel efficiency.

Another result is that changing the slip value in heterogeneous channels will be more pronounced at low Reynolds numbers and impact more. For example, when the slip coefficient changes from 0.025 to 0.1 in Figure 15, the mixing efficiency value decreases by 16%, while this reduced amount is roughly 6.4% for the same case in Figure 16. 

It can also be seen from Figure 16 that the mixing efficiency for Re = 0.009 and β > 0 reaches a constant value after approximately x > 3.5, and the change rate is less than 1%. In contrast, this change rate is about 7% for the same case for Re = 0.001 (Figure 15); therefore, it still requires more length to reach a constant value. On the other hand, it should be noted that this constant efficiency is achieved in a shorter length for low slip coefficients. For example, the constant mixing efficiency is obtained at x = 3.65 for Re = 0.009 and β = 0.025, while it is obtained at x = 3.85 (i.e., 5.5% longer length) for a similar Reynolds number and β = 0.05. This issue is important because the researchers and designers must know what range of Reynolds numbers, slip coefficients, and ultimately, the required channel length to consider in their micromixer designs to achieve a constant mixing efficiency. Similar results have been observed by the standard deviation of concentration in this field, confirming the accuracy of the above findings.

The results presented so far have been obtained for K=41 for the Debye–Huckel dimensionless parameter and the zeta-potential ratio $\zeta_{p}/\zeta_{m}=0.1$. It is worth noting that $\zeta_{p}$ and $\zeta_{m}$ relate to the size of the zeta-potential in the homogeneous and heterogeneous sections, respectively. Similar effects of increasing the Reynolds number are observed by changing the zeta-potential ratio in Figure 17. This figure shows that the mixing efficiency declines with increases in the zeta-potential ratio for β = 0.025. As can be inferred, the mass flow rate changes by changing the ratio of zeta-potential. Therefore, enhancing the mass flow rate reduces the mixing time and increases the flow speed passing through the microchannel.

In addition, decreasing the value of the Debye–Huckel parameter for the slip coefficient of 0.05 leads to improving the mixing efficiency in Figure 18. In fact, the mixing efficiency of hydrophobic surfaces is strongly affected by the thickness of the electric double layer; hence, reducing this thickness, which is inversely related to the Debye–Huckel value, improves the mixing efficiency. Decreasing the Debye–Huckel value means increasing the electric double layer thickness or the thickness of the region where the ions accumulate. This thickness increase means that the ions will have more free space and will not necessarily be close to the wall where the slip boundary condition is applied.

Therefore, the number of ions close to the surface, where the slip boundary condition is applied, decreases due to this thickness increase. In this case, the slip boundary condition will have a more negligible effect on the electrolyte and ions adjacent to the wall, and this will cause the fluid to spend more time in contact with the microchannel. As mentioned earlier, increasing β leads to facilitating the flow passing through the microchannel; in other words, increasing β causes the pumping effect to overcome the mixing effect. Hence, the fluid will move inside the channel at a lower speed by reducing the β effect.

## 5. Conclusions

The microfluidic applications for chemical and biological analysis have grown significantly in the last decade. One of the requirements for acquiring the technology of making such tools is a thorough understanding of related phenomena. This paper quantitatively studied the mixing of the electroosmotic flow inside a microchannel with a heterogeneous patch on the wall. The mixing in such an electroosmotic micromixer was analyzed by introducing appropriate criteria, and the crucial parameters were introduced to investigate the mixing rate. 

Electrokinetic mixing can be modeled through the H–S approximate model or completely solving the exact Nernst–Planck equations, which have great numerical solution difficulty. Mixing in the electroosmotic flow inside a microchannel with homogeneous walls is affected by molecular diffusion, which is not powerful. However, surface chemistry technology and creating heterogeneities in the electrical charges of the surface can produce significant rotational flows (vortices) inside the microchannel and improve the mixing efficiency. Numerical results confirm that breaking a large heterogeneity piece into several smaller heterogeneous patches and inserting them on the wall is a reasonable approach to better mixing efficiency in microchannels. As the heterogeneity patch position is closer to the microchannel inlet, the mixing performance is better. Therefore, the required length for favorable mixing inside the microchannels can be optimized after analyzing these findings. Using electroosmotic flow in a microchannel and correctly adjusting the heterogeneity patches of surface charges can design an electroosmotic micromixer with controllable mixing. 

It was also observed that the homogeneous channel is affected more by the Reynolds number so that its mixing efficiency will drastically decrease with increasing Reynolds numbers. In contrast, since there is less mixing efficiency variation for different Reynolds numbers and the same slip coefficient, we can conclude that the mixing efficiency change of heterogeneous microchannels is affected more by the slip coefficient than the Reynolds number. Therefore, we conclude that the usage of heterogeneous channels with hydrophobic surfaces does not provide satisfactory results for mixing in low Reynolds numbers, such that their mixing efficiency is even less than the homogeneous channel. In contrast, the presence of a low slip coefficient (e.g., β=0.025) at high Reynolds numbers on the heterogeneous microchannel wall will lead to better results and higher mixing efficiency than the regular homogeneous channel. It was also concluded that the role of slip boundary condition in heterogeneous channels is more pronounced at low Reynolds numbers. However, the mixing efficiency decreases with increasing slip coefficient in all Reynolds numbers because slip coefficient enhancement means that the pumping effects dominate the mixing effects, which reduces the effect of the vortices and their related rotational regions. These simple findings can be used to design and simulate complex electroosmotic micromixers having multiple heterogeneous patches.

## Figures and Tables

**Figure 1 micromachines-12-01055-f001:**
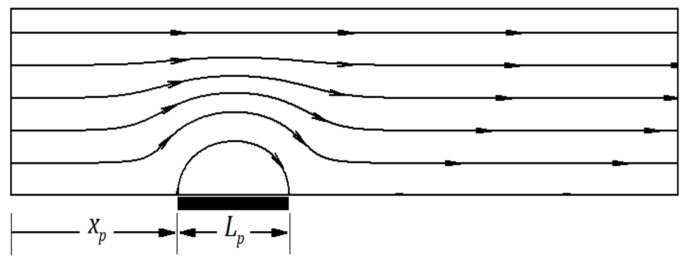
Streamlines related to the simplest heterogeneous microchannel design.

**Figure 2 micromachines-12-01055-f002:**
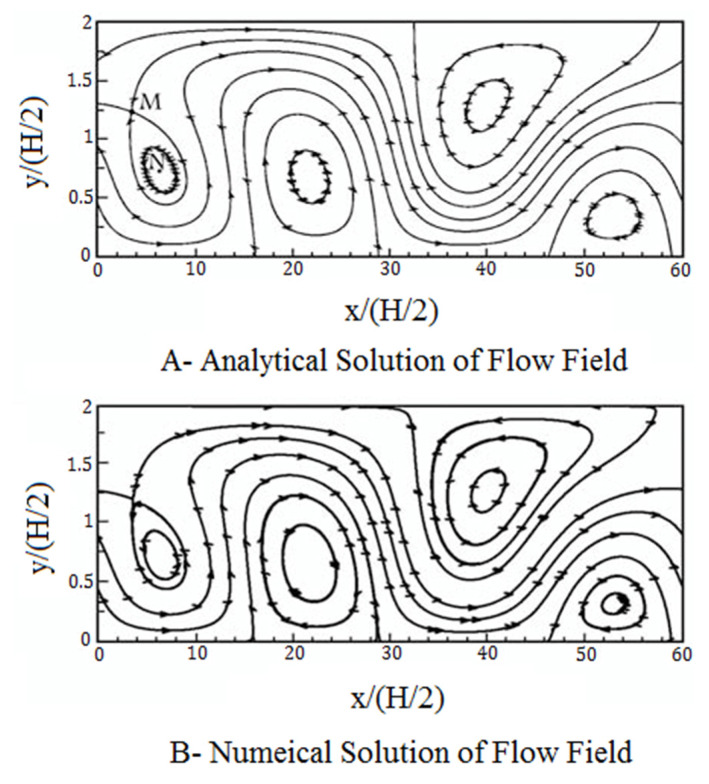
Streamlines obtained by (**A**). analytical solution and (**B**). numerical solution through the H–S approximate model.

**Figure 3 micromachines-12-01055-f003:**
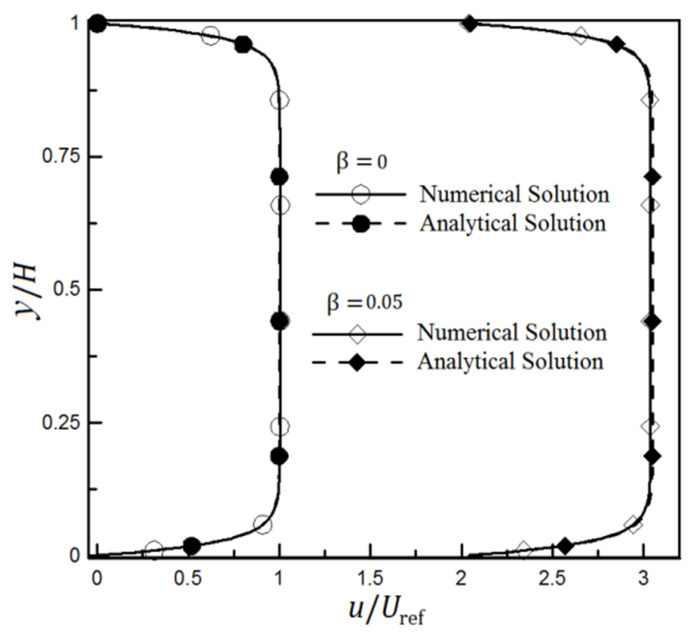
The velocity changes in a homogeneous microchannel through the full numerical solution model and analytical solution of electroosmotic velocity field with and without the slip boundary condition [34,62].

**Figure 4 micromachines-12-01055-f004:**
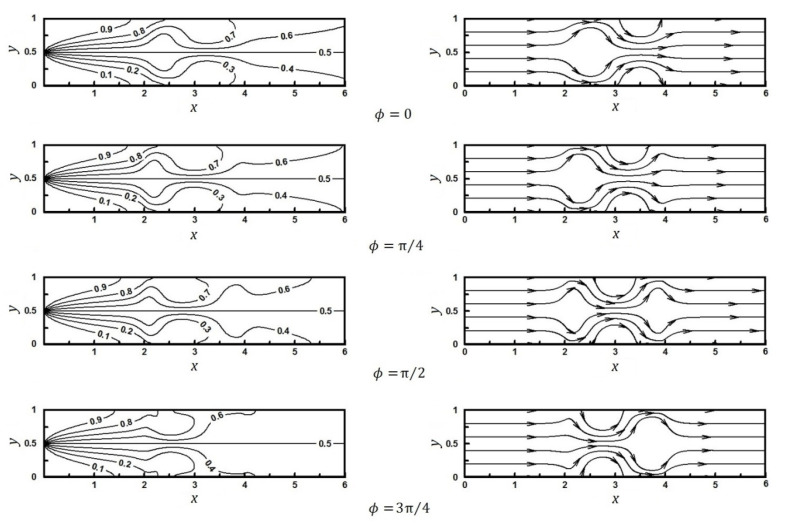
Flow fields related to Equation (20): Streamlines (right side) and concentration lines (left side).

**Figure 5 micromachines-12-01055-f005:**
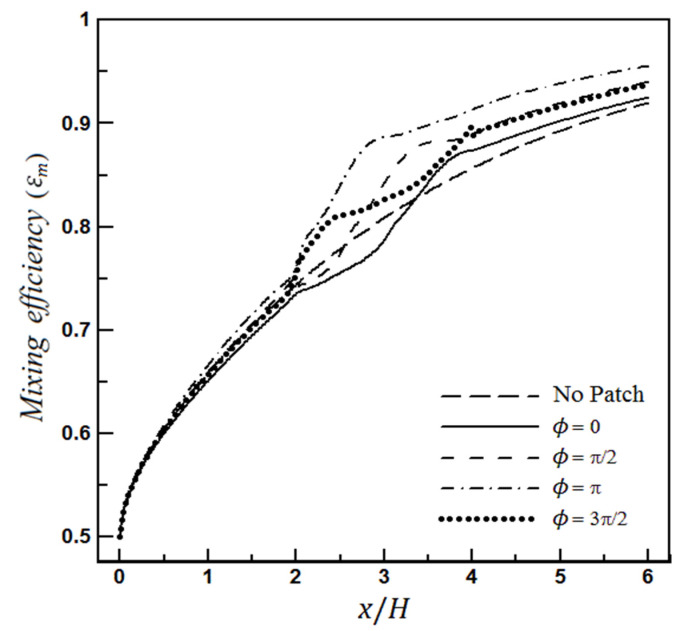
Mixing efficiency for flow fields of Equations (20) and (21).

**Figure 6 micromachines-12-01055-f006:**
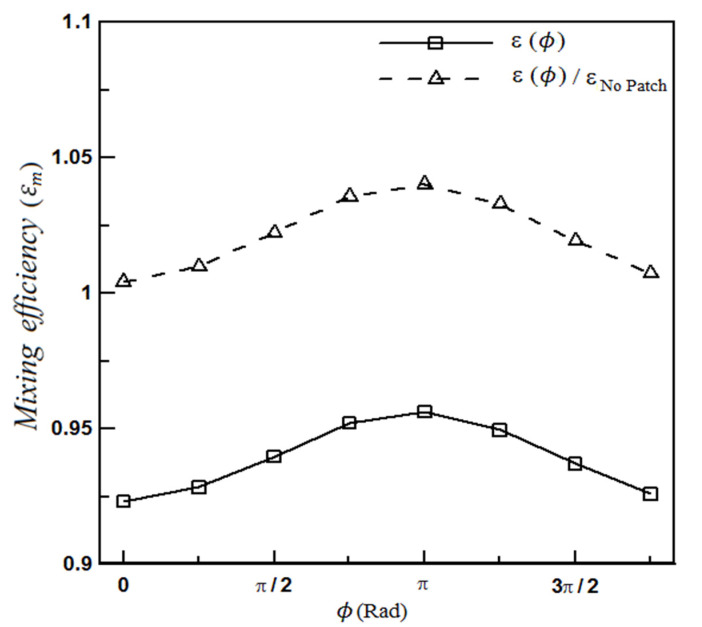
Relative and absolute mixing efficiency of microchannel for the flow fields of Equations (20) and (21).

**Figure 7 micromachines-12-01055-f007:**
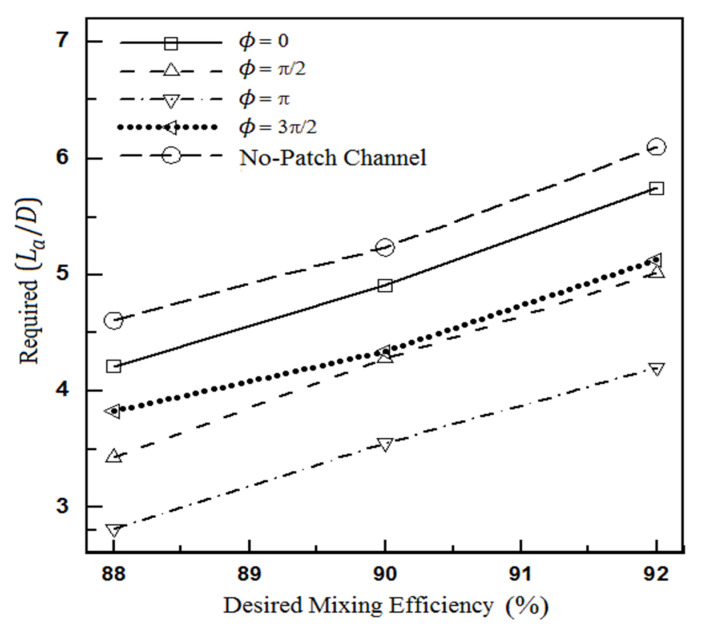
The length required to achieve the desired mixing efficiency for the flow fields of Equations (20) and (21).

**Figure 8 micromachines-12-01055-f008:**
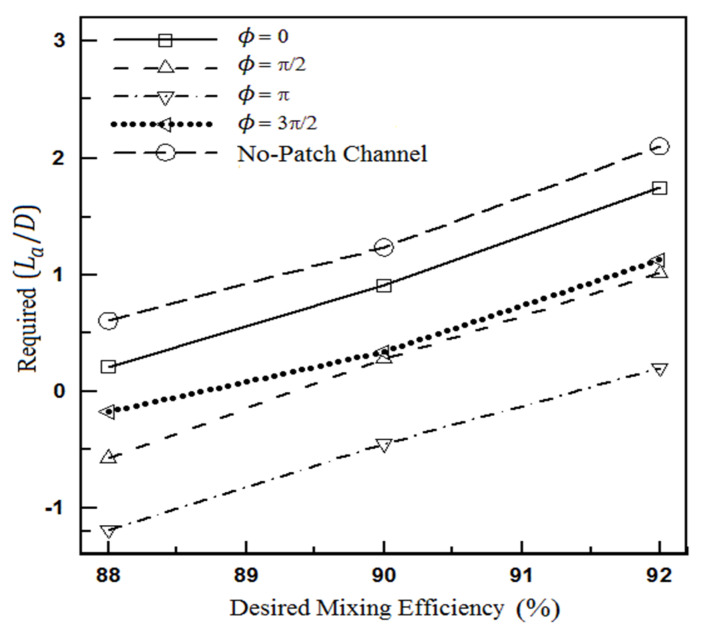
Additional length after heterogeneous section to achieve the desired mixing efficiency for the flow fields of Equations (20) and (21).

**Figure 9 micromachines-12-01055-f009:**
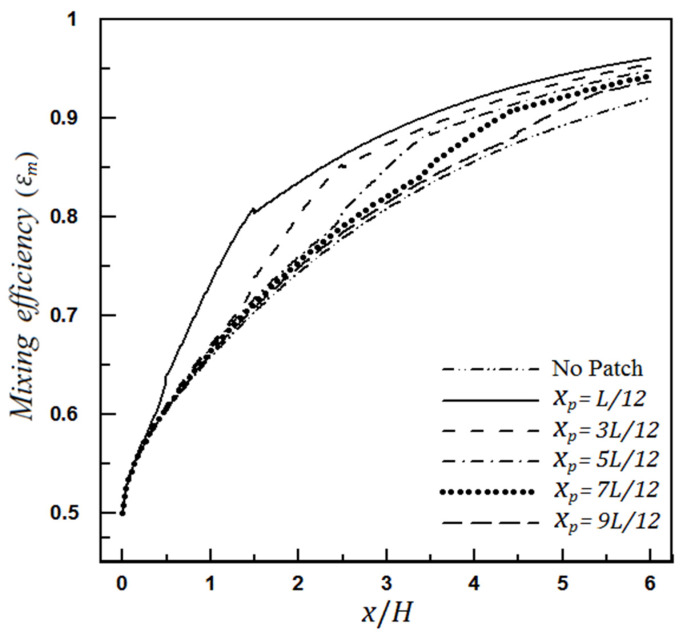
The effect of position changing of the heterogeneity patch on mixing efficiency (the length of heterogeneity patch, $L_{p}=L/6$).

**Figure 10 micromachines-12-01055-f010:**
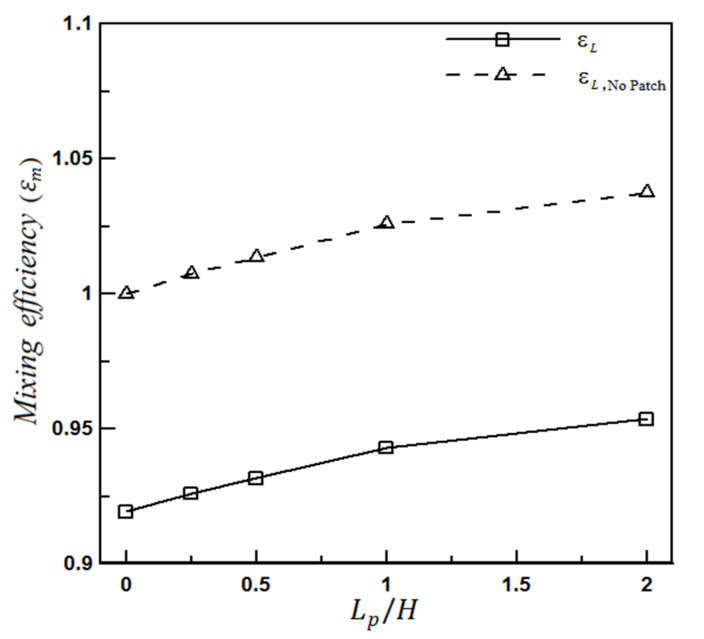
Effect of the heterogeneous patch size on mixing performance.

**Figure 11 micromachines-12-01055-f011:**
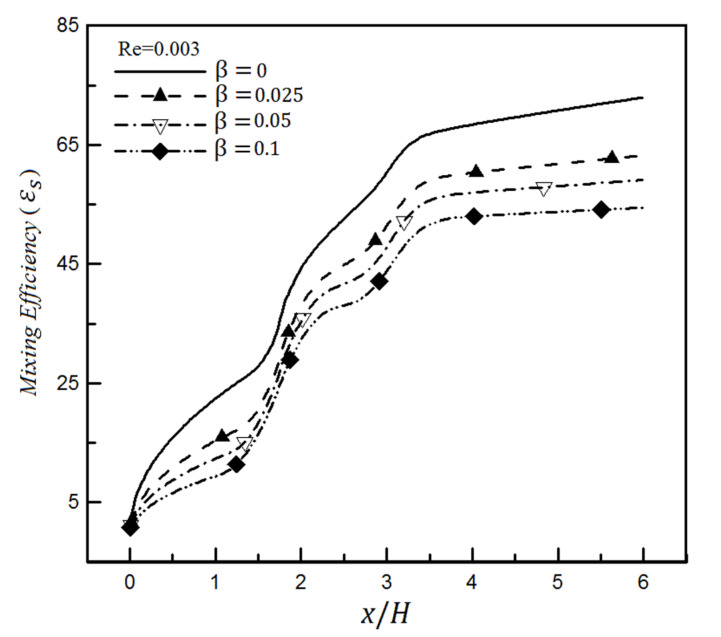
Mixing efficiency variations for different slip coefficients inside a microchannel having a heterogeneity patch based on mixing entropy index and Re = 0.003.

**Figure 12 micromachines-12-01055-f012:**
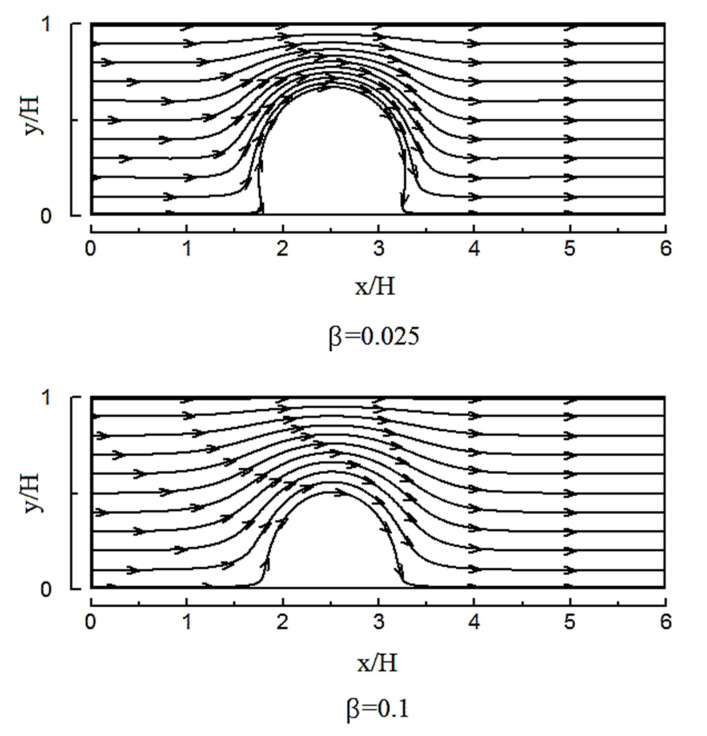
Streamlines for a heterogeneous microchannel with a heterogeneity patch for two slip coefficients of 0.025 and 0.1.

**Figure 13 micromachines-12-01055-f013:**
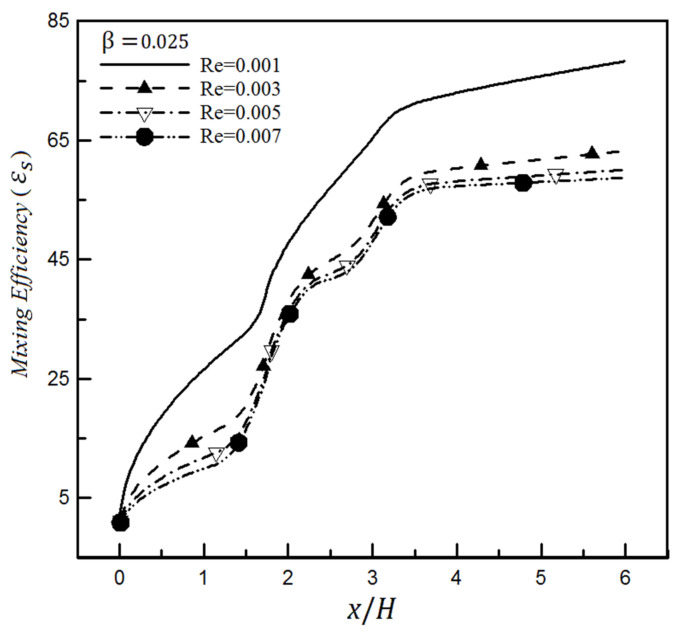
Mixing efficiency changes for a slip coefficient (0.025) and different Reynolds numbers through mixing entropy index.

**Figure 14 micromachines-12-01055-f014:**
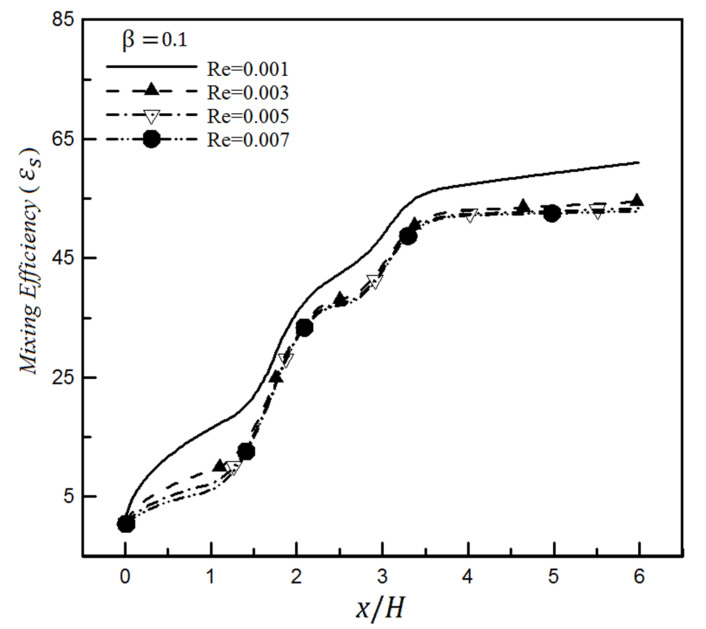
Mixing efficiency changes for a slip coefficient (0.1) and different Reynolds numbers through mixing entropy index.

**Figure 15 micromachines-12-01055-f015:**
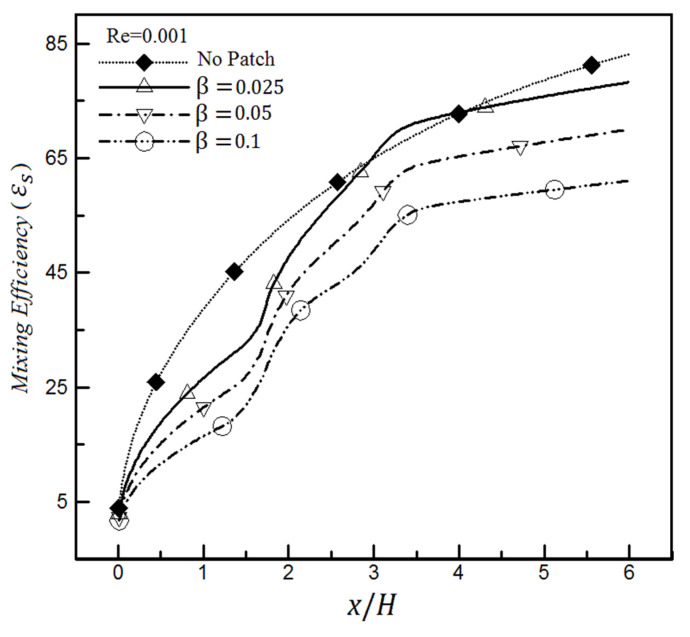
Mixing efficiency variations based on mixing entropy index in the non-homogeneous microchannel for various slip coefficients, Re = 0.001, and a comparison with the homogeneous channel.

**Figure 16 micromachines-12-01055-f016:**
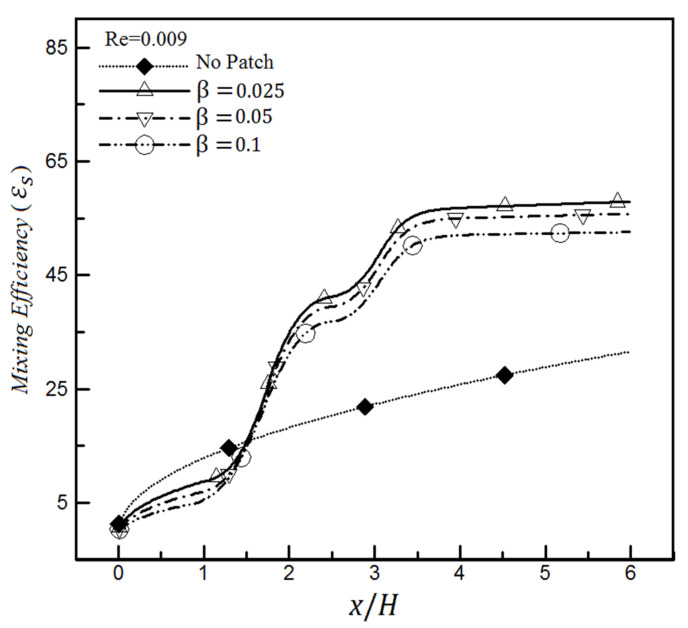
Mixing efficiency changes based on mixing entropy index inside the heterogeneous microchannel for different slip coefficients, Re = 0.009, and a comparison with the homogeneous channel.

**Figure 17 micromachines-12-01055-f017:**
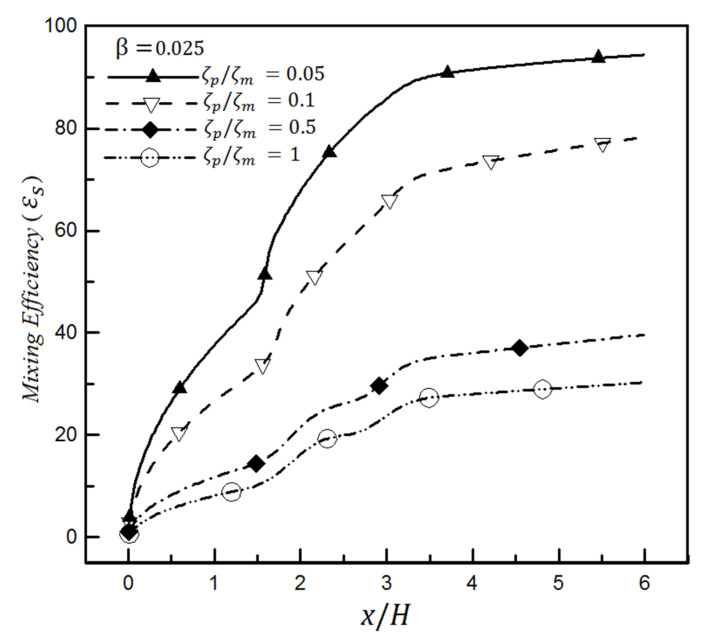
Mixing efficiency changes based on mixing entropy index in a heterogeneous microchannel for different zeta-potential ratios and a slip coefficient of 0.025.

**Figure 18 micromachines-12-01055-f018:**
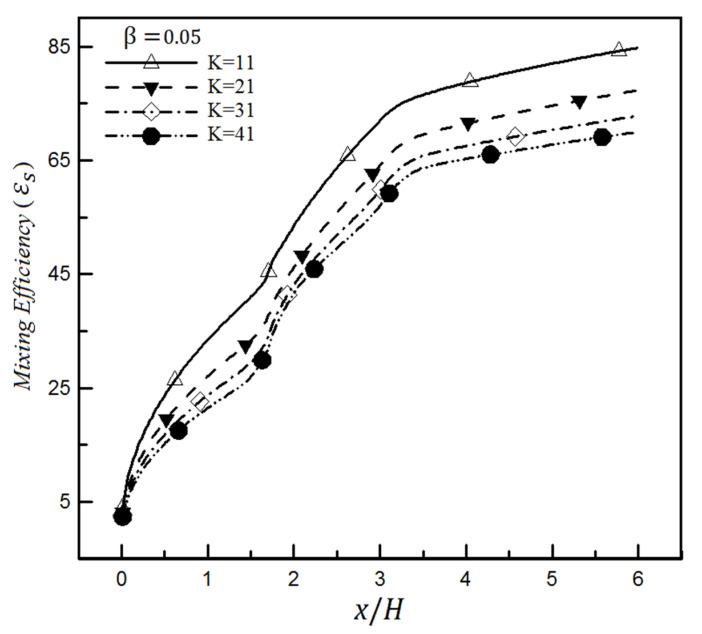
Mixing efficiency changes based on mixing entropy index in a heterogeneous microchannel for different Debye–Huckel values and a slip coefficient of 0.05.

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
