# Peer review of "Numerical Analysis of the Heterogeneity Effect on Electroosmotic Micromixers Based on the Standard Deviation of Concentration and Mixing Entropy Index"

_micromachines, 2021, doi:10.3390/mi12091055_

Round 1

Reviewer 1 Report

The authors present the numerical analysis of electroosmotic mixing with heterogeneity. There are some main comments on the manuscript: 

  1. The organization of the manuscript must be improved, especially on the mathematical description and the numerical scheme.
  2. The reviewer cannot see the novelty of the manuscript. The electroosmotic mixing has been studied extensively, even with complex geometry and full coupled model.
  3. The motivation of this work is not clear, such as why the authors choose to study the model. 
  4. A lot of information is missing in the manuscript, for example, the dimension of the geometry, the schematic diagram, several parameters. Combined with comment 1, the presentation of the manuscript is poor. 
  5. HS velocity is used, so the NS equation can be solved without considering the volumetric force,  but the numerical scheme solves for the potential. This does not make sense. 

Reviewer 2 Report

Minor proofreading is required.

Reviewer 3 Report

This paper presented a thorough work using numerical modeling methods to quantify the mixing effect cause by one heterogeneous patch in a microchanel of which the fluid driven by electro-osmotic flow, when the patch length is much larger than Debye length.

Although this idea wasn't new, the simulation results in details in this paper should be of high interests to microfluidic community. 

However, like what the authors also pointed out at the end of this paper, in real devices, there should be many patches in a microchannel, therefore the the ratio of patch length to spacing length has to be considered before using the results of this paper. 

The authors should address the aforementioned length ratio in the revision, which is quite important as far as the usefulness of this work is concerned.  

Round 2

Reviewer 1 Report

The authors have addressed the comments properly.